# WHERE AND WHEN TO LOOK? SPATIO-TEMPORAL ATTENTION FOR ACTION RECOGNITION IN VIDEOS

## ABSTRACT

Inspired by the observation that humans are able to process videos efficiently by only paying attention *where* and *when* it is needed, we propose a novel spatial-temporal attention mechanism for video-based action recognition. For spatial attention, we learn a saliency mask only using convolutional layers to allow the model to focus on the most salient parts of the feature maps. For temporal attention, we employ a convolutional LSTM based attention mechanism to identify the most relevant frames from an input video. Further, we propose a set of regularizers that ensure that our attention mechanism attends to coherent regions in space and time. Our model can not only effectively improve video action recognition accuracy, but also can localize discriminative regions both spatially and temporally, despite only trained in a weakly-supervised manner with only classification labels (no bounding box spatial labels and time frame temporal labels). We evaluate our proposed approach on several public video action recognition datasets with ablation studies. Furthermore, we quantitatively and qualitatively evaluate our model's ability to localize discriminative regions spatially and critical frames temporally. Experimental results demonstrate the efficacy of our approach, showing superior or comparable accuracy with the state-of-the-art methods with the same input.

## 1 INTRODUCTION

An important property of human perception is that one does not need to process a whole scene in its entirety at once. Instead, humans focus attention selectively on parts of the visual space to acquire information *where* and *when* it is needed, and combine information from different fixations over time to build up an internal representation of the scene (Rensink, 2000), which can then be used for interpretation or decision making.

In computer vision and natural language processing, over the last couple of years, attention models have proved similarly important. Particularly for the tasks where interpretation or explanation requires only a small portion of the image or video. Examples include visual question answering (Lu et al., 2016; Xu & Saenko, 2016; Xiong et al., 2016), activity recognition (Sharma et al., 2015; Girdhar & Ramanan, 2017; Li et al., 2018b), and natural machine translation (Bahdanau et al., 2015). These models have also provided a level of interpretability, by visualizing regions selected or attended over for a particular task or decision. In particular, for video action classification, a proper attention model can help answer the question of *where* and *when* it needs to look at the image evidence to draw a classification decision. It intuitively explains which part the model attends to when making a particular decision, which is very helpful in real applications, e.g., medical AI systems or self-driving cars.

In this paper, we propose a novel spatio-temporal attention mechanism that is designed to address these challenges. Our attention mechanism is efficient, due to its space- and time- separability, and yet flexible enough to enable encoding of effective regularizers (or priors). As such, our attention mechanism consists of spatial and temporal components shown in Fig. 1. The spatial attention component, that attenuates frame-wise CNN image features, consists of the saliency mask; regularized to be discriminative and spatially smooth. The temporal component consists of a uni-modal soft attention mechanism that aggregates information over the near-by attenuated frame features before passing it into Convolutional LSTM for class prediction.

**Contributions:** In summary, the main contributions of this work are: (1) We introduce a simple yet effective spatial-temporal attention for video action recognition, which consists of the saliency mask

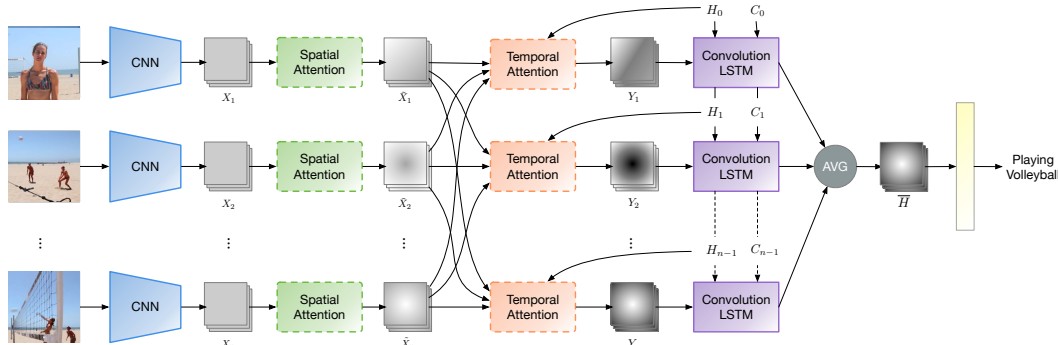

Figure 1: **Spatio-temporal attention for video action recognition.** The convolutional features are attended over both spatially, in each frame, and subsequently temporally. Both attentions are soft, meaning that the effective final representation at time $t$ of an RNN, used to make the prediction, is a spatio-temporally weighted aggregation of convolutional features across the video along with the past hidden state from $t - 1$. For details please refer to Sec. 3.

for spatial attention learned by ConvNets and temporal attention learned by convolutional LSTM. (2) We introduce three different regularizers, two for spatial and one for temporal attention components, to improve performance and interpretability of our model; (3) We demonstrate the efficacy of our model for video action recognition in three public datasets and explore the importance of our modeling choices through ablation experiments; (4) Finally, we qualitatively and quantitatively show that our spatio-temporal attention is able to localize discriminative regions and important frames, despite being trained in a purely weakly-supervised manner with only classification labels.

## 2 RELATED WORK

### 2.1 NETWORK INTERPRETATION

Various methods have been proposed to try to explain neural networks (Zeiler & Fergus, 2014; Springenberg et al., 2014; Mahendran & Vedaldi, 2016; Zhou et al., 2016; Zhang et al., 2016; Simonyan et al., 2013; Ramprasaath et al., 2016; Ribeiro et al., 2016; 2018; Chang et al., 2018) in various of ways, including visualizing the gradients, perturbing the inputs, and bridging relations with other well-studied systems. Visual attention is also one way that tries to explain which part of the image is responsible for the network's decision (Li et al., 2018a; Jetley et al., 2018). Besides the explanation, Li et al. (2018a) build up an end-to-end model to provide supervision directly on these explanations, specifically network's attention.

### 2.2 VISUAL ATTENTION FOR VIDEO ACTION RECOGNITION

For video action recognition, visualizing which part of the frame and which frame of the video sequence that the model was attending to provides valuable insight into the model's behavior. Sharma et al. (2015) develop an attention-driven LSTM by highlighting important spatial locations for action recognition. Girdhar & Ramanan (2017) introduce an attention mechanism based on a derivation of bottom-up and top-down attention as low-rank approximations of bilinear pooling methods. However, these work only focus on the crucial spatial locations of each image, without considering temporal relations among different frames in a video sequence. To alleviate this shortcoming, visual attention is incorporated in the motion stream (Wang et al., 2016b; Li et al., 2018b; Du et al., 2018). However, the motion stream only employs the optical flow frames generated from consequent frames, cannot consider the long-term temporal relations among different frames in a video sequence. Moreover, motion stream needs additional optical flow frames as input, which imposes burden due to additional optical flow extraction, storage and computation and is especially severe for large datasets. Torabi & Sigal (2017) propose an attention based LSTM model to hightlight frames in videos, but spatial information is not used for temporal attention. An end-to-end spatial

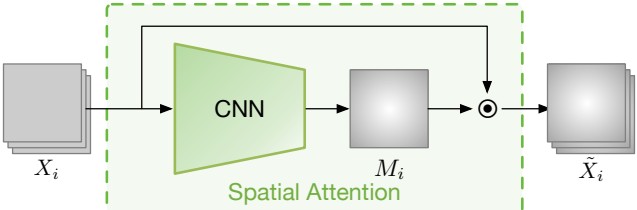

Figure 2: **Spatial attention component.** We use several layers of convolutional network to learn the importance mask $M_i$ for the input image feature $X_i$, the output is the element-wise multiplication $\tilde{X}_i = X_i \odot M_i$. Details please refer to Sec. 3.2.

and temporal attention model is proposed in (Song et al., 2017) for human action recognition, but additional skeleton data is needed.

## 3 Spatial-Temporal Attention Mechanism

Our overall model is an Recurrent Neural Network (RNN) that aggregates frame-based convolutional features across the video to make action predictions as shown in Fig. 1. The convolutional features are attended over both spatially, in each frame, and subsequently temporally. Both attentions are soft, meaning that the effective final representation at time $t$ of an RNN, used to make the prediction, is a spatio-temporally weighted aggregation of convolutional features across the video along with the past hidden state from $t - 1$. The core novelty is the overall form of our attention mechanism and the additional terms of the loss function that induce sensible spatial and temporal attention priors.

### 3.1 Convolutional Frame Features

We use the last convolutional layer output extracted by ResNet50 or ResNet101 (He et al., 2016), pretrained on the ImageNet (Deng et al., 2009) dataset and fine-tuned for the target dataset, as our frame feature representation. We acknowledge that more accurate feature extractors (for instance, network with more parameters such as ResNet-152 or higher performance networks such as DenseNet (Huang et al., 2017) or SENet (Hu et al., 2018)) and optical flow features will likely lead to better overall performance. Our primary purpose in this paper is to prove the efficacy of our spatial-temporal attention mechanism. Hence we kept the features relatively simple.

### 3.2 Spatial Attention With Importance Mask

We apply an importance mask $M_i$ to the $i$-th image features $X_i$ to obtain attended image features by element-wise multiplication:

$$\tilde{X}_i = X_i \odot M_i, \tag{1}$$

for $1 \leq i \leq n$. This operation attenuates certain regions of the feature map based on their estimated importance. Here we simply use three convolutional layers to learn the importance mask (please refer to Appendix B.2 for network architecture details). Fig. 2 illustrates our spatial attention mechanism. However, if left uncontrolled, an arbitrarily structured mask could be learned, leading to possible overfitting. We posit that, in practice, it is often useful to attend to a few important larger regions (e.g., objects, elements of the scene). To induce this behavior, we encourage smoothness of the mask by introducing total variation loss on the spatial attention, as will be described in Sec. 3.4.

### 3.3 Temporal Attention

Inspired by attention for neural machine translation (Bahdanau et al., 2015), we introduce the temporal attention mechanism which generates *energy* for each attended frame $\tilde{X}_i$ at each time step $t$,

$$e_{ti} = \Phi(H_{t-1}, \tilde{X}_i), \tag{2}$$

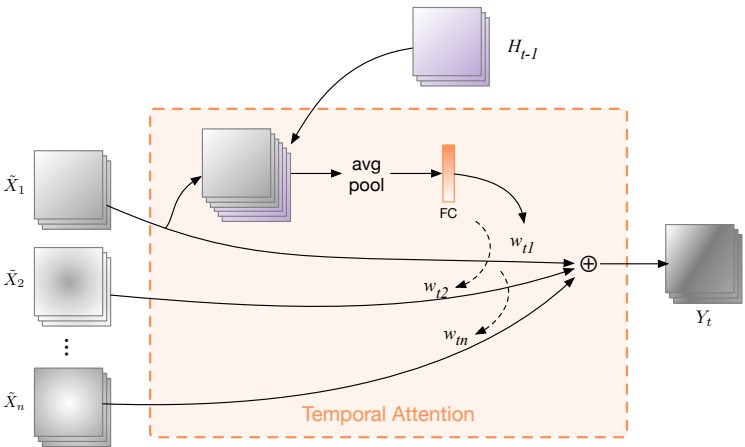

Figure 3: **Temporal attention component.** The temporal attention learns a temporal attention weight $w_{ti}$ at each time step $t$. The final feature map $\boldsymbol{Y}_t$ at time $t$ to the ConvLSTM is a weighted sum of the feature from all the previous masked frames. Details please refer to Sec. 3.3.

where $\boldsymbol{H}_{t-1}$ represents the ConvLSTM hidden state at time $t-1$ that implicitly contains all previous information up to time step $t-1$, $\tilde{\boldsymbol{X}}_i$ represents the $i$-th frame masked features. $\Phi = \Phi_H(\boldsymbol{H}_{t-1}) + \Phi_X(\tilde{\boldsymbol{X}}_i)$, where $\Phi_H$ and $\Phi_X$ are feed-forward neural networks which are jointly trained with all other components of the proposed system.

This temporal attention model directly computes soft attention weight for each frame at each time $t$ as shown in Fig. 3. It allows the gradient of the cost function to be backpropagated through. This gradient can be used to train the entire spatial-temporal attention model jointly.

The importance weight $w_{ti}$ for each frame is:

$$w_{ti} = \frac{\exp(e_{ti})}{\sum_{i=1}^{n}(\exp(e_{ti}))} \tag{3}$$

for $1 \leq i \leq n, 1 \leq t \leq T$. This importance weighting mechanism decides which frame of the video to pay attention to. The final feature map $\boldsymbol{Y}_t$ to the ConvLSTM is a weighted sum of the feature from all of the frames as the ConvLSTM cell inputs:

$$\boldsymbol{Y}_t = \frac{1}{n}\sum_{i=1}^{n} w_{ti}\tilde{\boldsymbol{X}}_i \tag{4}$$

where $\tilde{\boldsymbol{X}}_i$ denotes the $i$-th masked frame of each video, $n$ represents the total number of frames for each video.

For RNN, instead of using conventional LSTM (Graves, 2013), we use Convolutional LSTM (ConvLSTM) (Shi et al., 2015) instead. The drawback of conventional LSTM is its use of full connections in the input-to-state and state-to-state transitions in which no spatial information is encoded. In contrast, each input, cell output, hidden state, gate are 3D tensors whose last two dimensions are spatial dimensions which can preserve spatial information, which is more suitable for image inputs.

We use the following initialization strategy for the ConvLSTM cell state and hidden state for faster convergence:

$$\boldsymbol{C}_0 = g_c(\frac{1}{n}\sum_{i=1}^{n}\tilde{\boldsymbol{X}}_i), \qquad \boldsymbol{H}_0 = g_h(\frac{1}{n}\sum_{i=1}^{n}\tilde{\boldsymbol{X}}_i) \tag{5}$$

where $g_c$ and $g_h$ are two layer convolutional networks with batch normalization (Ioffe & Szegedy, 2015).

We calculate the average hidden states of ConvLSTM over time length $T$,

$$\overline{\boldsymbol{H}} = \frac{1}{T}\sum_{i=1}^{T}\boldsymbol{H}_i \tag{6}$$

and send it to a fully connected classification layer for the final video action classification.

### 3.4 LOSS FUNCTION

Considering the spatial and temporal nature of our video action recognition; we would like to learn (1) a sensible attention mask for spatial attention, (2) reasonable importance weighting scores for different frames, and (3) improve the action recognition accuracy at the same time. Therefore, our loss function $L$:

$$L = L_{\text{CE}} + \lambda_{\text{TV}} L_{\text{TV}} + \lambda_{\text{contrast}} L_{\text{contrast}} + \lambda_{\text{unimodal}} L_{\text{unimodal}}, \quad (7)$$

where $L_{\text{CE}}$ is the cross entropy loss for classification, $L_{\text{TV}}$ represents the total variation regularization (Rudin et al., 1992); $L_{\text{contrast}}$ represents the mask and background contrast regularizer; and $L_{\text{unimodal}}$ represents unimodality regularizer. $\lambda_{\text{TV}}$, $\lambda_{\text{contrast}}$ and $\lambda_{\text{unimodal}}$ are the weights for corresponding regularizers.

The total variation regularization $L_{\text{TV}}$ of the learnable attention mask encourages spatial smoothness of the mask and is defined as:

$$L_{\text{TV}} = \sum_{i=1}^{n} \left( \sum_{j,k} |\boldsymbol{M}_i^{j+1,k} - \boldsymbol{M}_i^{j,k}| + \sum_{j,k} |\boldsymbol{M}_i^{j,k+1} - \boldsymbol{M}_i^{j,k}| \right) \quad (8)$$

where $\boldsymbol{M}_i$ is the mask for the $i$-th frame, and $\boldsymbol{M}_i^{j,k}$ is entry at the $(j,k)$-th spatial location of the mask. Different from the total variation of the mask of using $L_2$ loss in Dabkowski & Gal (2017), we use $L_1$ loss instead. The contrast regularization $L_{\text{contrast}}$ of learnable attention mask is to suppress the irrelevant information and highlight important information:

$$L_{\text{contrast}} = \sum_{i=1}^{n} \left( -\frac{1}{2} \boldsymbol{M}_i \odot \boldsymbol{B}_i + \frac{1}{2} \boldsymbol{M}_i \odot (1 - \boldsymbol{B}_i) \right) \quad (9)$$

where $\boldsymbol{B}_i = \mathbb{I}\{\boldsymbol{M}_i > 0.5\}$ represents the binarized mask, $\mathbb{I}$ is the indicator function applied element-wise.

The unimodality regularizer $L_{\text{unimodal}}$ encourages the temporal attention weights to be unimodal, biasing against spurious temporal weights. This stems from our observation that in most cases only one activity would be present in the considered frame window, with possible irrelevant information on either or both sides. Here we use the log concave distribution to encourage the unimodal pattern of temporal attention weights:

$$L_{\text{unimodal}} = \sum_{t=1}^{T} \sum_{i=2}^{n-1} \max\{0, w_{t,i-1} w_{t,i+1} - w_{t,i}^2\} \quad (10)$$

where $T$ represents the ConvLSTM time sequence length and $n$ is the number of frames for each video. More details on this log concave sequence please refer to Appendix A.

## 4 EXPERIMENTS

In this section, we first conduct experiments to evaluate our proposed method on video action recognition task on three public available datasets. Then we evaluate our spatial attention mechanism on the spatial localization task and our temporal attention mechanism on the temporal localization task respectively.

### 4.1 VIDEO ACTION RECOGNITION

We first conduct extensive studies on the widely used HMDB51 and UCF101 datasets. The purpose of these experiments is mainly for ablation study to examine the effects of different sub-components. Then we show that our method can be applied to the challenging large-scale Moments in Time dataset.

**Datasets.** HMDB51 dataset (Kuehne et al., 2011) contains 51 distinct action categories, each containing at least 101 clips for a total of 6,766 video clips extracted from a wide range of sources. These videos include general facial actions, general body movements, body movements with object interaction, body movements for human interaction.

| Model | HMDB51 | UCF101 |
|---|---|---|
| Visual attention (Sharma et al., 2015) | 41.31 | 84.96 |
| VideoLSTM (Li et al., 2018b) | 43.30 | 79.60 |
| Attentional Pooling (Girdhar & Ramanan, 2017) | 50.80 | – |
| ResNet101-ImageNet | 50.04 | 83.30 |
| **Ours** | **53.07** | **87.11** |
| **Ablation Experiments** | | |
| Ours w/o spatial attentionz | 51.98 | 85.78 |
| Ours w/o temporal attention | 52.25 | 85.86 |
| Ours w/o $L_{TV}$ | 52.01 | 85.89 |
| Ours w/o $L_{contrast}$ | 52.10 | 85.98 |
| Ours w/o $L_{unimodal}$ | 52.05 | 86.10 |

Table 1: **Top-1 accuracy (%) on HMDB51 and UCF101 dataset.**

UCF101 dataset (Soomro et al., 2012) is an action recognition dataset of realistic action videos, collected from YouTube, with 101 action categories.

Moments in Time Dataset (Monfort et al., 2018) is a collection of one million short videos with one action label per video and 339 different action classes. As there could be more than one action taking place in a video, action recognition models may predict an action correctly yet be penalized because the ground truth does not include that action. Therefore, it is believed that top 5 accuracy measure will be more meaningful for this dataset.

**Experimental setup.** We use the same parameters for HMDB51 and UCF101: single Convolutional LSTM layer with hidden-state dimension 512, sequence length $T = 25$, $\lambda_{TV} = 10^{-5}$, $\lambda_{contrast} = 10^{-4}$, $\lambda_{unimodal} = 1$. For the Moments in Time dataset, we use time sequence length $T = 15$. For more details on the experimental setup please refer to Appendix B.1.

**Quantitative results.** We show the top-1 video action classification accuracy for HMDB51 and UCF101 dataset in Table 1. Our proposed model outperforms previous attention based model (Sharma et al., 2015; Li et al., 2018b; Girdhar & Ramanan, 2017) and conventional ResNet101-ImageNet. From the ablation experiments, it demonstrates that all the sub-components of the proposed method contribute to improving the final performance.

The results on the Moments in Time dataset are reported in Table 2. Our method achieves the best accuracy comparing to other single-modality-based methods, and obtains better or comparative results comparing to the methods which uses more than one modality. TRN-Multiscale (Zhou et al., 2018), which uses both RGB and optical flow images, has better performance than ours, however, extracting optical flow images for such large datasets is very time-consuming and needs the same order of magnitude of storage as RGB images.

| Model | Modality | Top-1 (%) | Top-5 (%) |
|---|---|---|---|
| ResNet50-ImageNet (Monfort et al., 2018) | RGB | 26.98 | 51.74 |
| TSN-Spatial (Wang et al., 2016a) | RGB | 24.11 | 49.10 |
| TRN-Multiscale (Zhou et al., 2018) | RGB | 27.20 | 53.05 |
| BNInception-Flow (Monfort et al., 2018) | Optical flow | 11.60 | 27.40 |
| ResNet50-DyImg (Monfort et al., 2018) | Optical flow | 15.76 | 35.69 |
| TSN-Flow (Wang et al., 2016a) | Optical flow | 15.71 | 34.65 |
| TSN-2stream (Wang et al., 2016a) | RGB+Optical flow | 25.32 | 50.10 |
| TRN-Multiscale (Zhou et al., 2018) | RGB+optical flow | **28.27** | **53.87** |
| **Ours** | RGB | 27.55 | 53.52 |

Table 2: **Results on Moments in Time dataset.** ResNet50-ImageNet and TRN-Multiscale spatial results reported here are based on authors' (Monfort et al., 2018) released trained model.

**Qualitative results.** We visualize the spatial attention and temporal attention results in Fig. 4. We can see that the spatial attention can correctly focus on important spatial area of the image, and the temporal attention shows a unimodal distribution for the entire action from starting the action to completing the action. More results are shown in Appendix C.1.

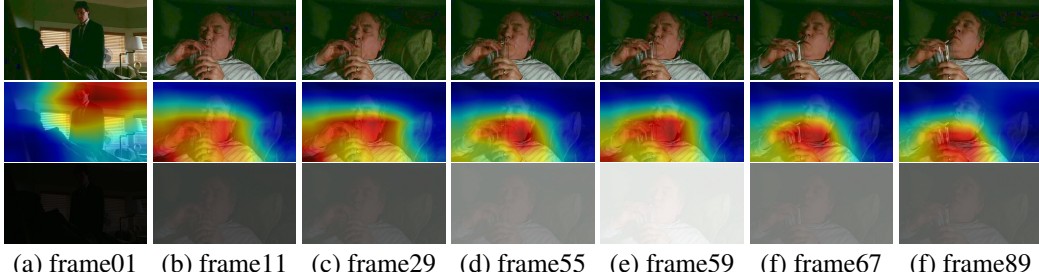

(a) frame01   (b) frame11   (c) frame29   (d) frame55   (e) frame59   (f) frame67   (f) frame89

Figure 4: **Examples of spatial temporal attention.** (Best viewed in color.) A frame sequence from a video of *Drink* action in HMDB51. The original images are shown at the top row, spatial attention is shown as heatmap (red means important) in the middle row, and temporal attention score is shown as the gray image (the brighter the frame is, the more crucial the frame is) at the bottom row. It shows that spatial attention can focus on important areas while temporal attention can attend to crucial frames. The temporal attention also shows a unimodal distribution for the entire action from starting to drink to completing the action.

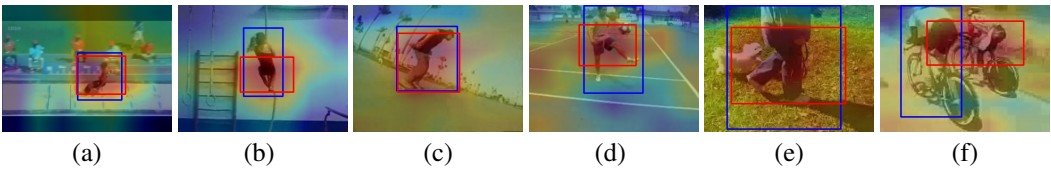

(a)      (b)      (c)      (d)      (e)      (f)

Figure 5: **Examples of spatial attention for action localization.** (Best viewed in color.) Blue bounding boxes represent ground truth while the red ones are predictions from our learned spatial attention. (a) long jump, (b) rope climbing, (c) skate boarding, (d) soccer juggling, (e) walking with dog, (f) biking.

## 4.2 WEAKLY SUPERVISED LOCALIZATION

Due to the existence of spatial and temporal attention mechanisms, our model can not only classify the action of the video, but also give a better interpretability of the results, *i.e.* telling which region and frames contribute more to the prediction. In other words, our proposed model can also localize the most discriminant region and frames at the same time. To verify this, we conduct the spatial localization and temporal localization experiments.

### 4.2.1 SPATIAL ACTION LOCALIZATION

**Dataset.** UCF101-24 is a subset of 24 classes out of 101 classes of UCF101 that comes with spatio-temporal localization annotation, released as bounding box annotations of humans with THU-MOS2013 and THUMOS2014 challenge (Jiang et al., 2014).

**Experimental setup.** For training, we only use the classification labels without spatial bounding box labels. For evaluation, we threshold the produced saliency mask at 0.5 and the tightest bounding box that contains the thresholded saliency map is set as the predicted localization box for each frame. Then these predicted localization boxes are compared with the ground truth bounding boxes at different Intersection Over Union (IOU) levels.

**Qualitative results.** We show some qualitative results in Fig. 5. Our spatial attention can attend to important action areas. The ground truth bounding boxes include all the entire human actions, while our attention could attend to crucial parts of an action such as in Fig.5 (d) and (e). Furthermore, our attention mechanism is able to attend to areas with multiple human actions. For instance, in Fig.5 (f) the ground truth only includes one person bicycling, but our attention can include both people bicycling. More qualitative results including failure cases are included in Appendix C.2.

**Quantitative results.** Table 3 shows the quantitative results for UCF101-24 spatial localization results. Our attention mechanism works better compared with the baseline methods when the IoU threshold is lower mainly because our model only focuses on important spatial areas rather than the entire human action annotated by bounding boxes. Compared with the baseline methods train-

| Methods | $\alpha = 0.05$ | $\alpha = 0.1$ | $\alpha = 0.2$ | $\alpha = 0.3$ |
|---|---|---|---|---|
| **Fast action proposal *** (Yu & Yuan, 2015) | 42.8% | – | – | – |
| **Learning to track *** (Weinzaepfel et al., 2015) | 54.3% | 51.7% | 47.7% | 37.8% |
| **R-C3D *** (Xu et al., 2017) | – | 54.4% | **51.5**% | **44.8**% |
| **Ours** | **67.0**% | **56.1**% | 34.1% | 17.7% |

Table 3: **Spatial action localization results on UCF101-24 dataset** measured by mAP at different IoU thresholds $\alpha$. * The baseline methods are strongly supervised spatial localization methods.

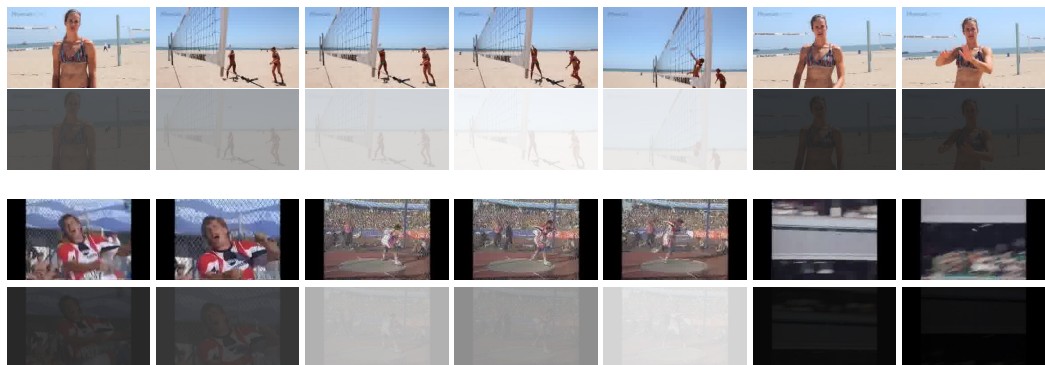

Figure 6: **Examples temporal localization with temporal attention from THUMOS14.** The upper two rows show *Volleyball* action original images and imposed with temporal attention weights respectively. The lower two rows show *Throw Discus* action. Our temporal attention module can automatically highlight important frames and avoid irrelevant frames corresponding to non-action poses or background.

ing with ground truth bounding boxes, we only use the action classification label, no ground truth bounding boxes are used.

### 4.2.2 TEMPORAL ACTION LOCALIZATION

**Dataset.** The action detection task of THUMOS14 (Jiang et al., 2014) consists of 20 classes of sports activities, and contains 2765 trimmed videos for training, while 200 and 213 untrimmed videos for validation and test respectively. More details on this dataset and pre-processing are included in Appendix B.1.

**Experimental setup.** We use the same hyperparameters for THUMOS14 as HMDB51, UCF101 and UCF101-24. For training, we only use the classification labels without temporal annotation labels. For evaluation, we threshold the normalized temporal attention importance weight at 0.5. Then these predicted temporal localization frames are compared with the ground truth annotation at different IoU thresholds.

**Qualitative results.** We first visualize some examples of learned attention weights on the test data of THUMOS14 in Fig. 6. We see that our temporal attention module is able to automatically highlight important frames and to avoid irrelevant frames corresponding to background or non-action human poses. More qualitative results are included in Appendix C.4.

**Quantitative results.** With our spatial temporal attention mechanism, the video action classification accuracy for the THUMOS'14 20 classes improved from 74.45% to 78.33%: a 3.88% increase. Besides improving the classification accuracy, we show our temporal attention mechanism is able to highlight discriminative frames quantitatively in Table 4. Compared with reinforcement learning based method (Yeung et al., 2016) and weakly supervised method (Wang et al., 2017), our method achieves the best accuracy in terms of different levels of IoU thresholds.

| Method | $\alpha = 0.1$ | $\alpha = 0.2$ | $\alpha = 0.3$ | $\alpha = 0.4$ | $\alpha = 0.5$ |
|---|---|---|---|---|---|
| (Yeung et al., 2016) | 48.9% | 44.0% | 36.0% | 26.4% | 17.1% |
| (Wang et al., 2017) | 44.4% | 37.7% | 28.2% | 21.1% | 13.7% |
| **Ours** | **70.0%** | **61.4%** | **48.6%** | **32.6%** | **17.9%** |

Table 4: **Temporal action localization results on THUMOS'14 dataset** measured by mAP at different IoU thresholds $\alpha$.

## 5 CONCLUSION

In this work, we develop a novel spatial-temporal attention mechanism for the task of video action recognition, demonstrating the efficacy across three publicly available datasets. Also, we introduce a set of regularizers that ensure our attention mechanism attends to coherent regions in space and time, further improving the performance and increasing the model interpretability. Moreover, we qualitatively and quantitatively show that our spatio-temporal attention is able to localize discriminative regions and important frames, despite being trained in a purely weakly-supervised manner with only classification labels.

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

## A    LOG-CONCAVE SEQUENCE

In probability and statistics, a unimodal distribution is a probability distribution that has a single peak or mode. If a distribution has more modes it is called multimodal. The temporal attention weights are a univariate discrete distribution over the frames, indicating the importance of the frames for the task of classification. In the context of activity recognition, it is reasonable to assume that the frames that contain salient information should be consecutive, instead of scattered around. Therefore, we would like to design a regularizer that encourages unimodality. To this end, we introduce a mathematical concept called the log-concave sequence and define the regularizer based on it.

We first give a formal definition of the unimodal sequence.

**Definition 1.** *A sequence $\{a_i\}_{i=1}^n$ is unimodal if for some integer m,*

$$\begin{cases} a_{i-1} \leq a_i & \text{if } i \leq m, \\ a_i \geq a_{i+1} & \text{if } i \geq m. \end{cases}$$

A univariate discrete distribution is unimodal, if its probability mass function forms a unimodal sequence. The log-concave sequence is defined as follows.

**Definition 2.** *A non-negative sequence $\{a_i\}_{i=1}^n$ is log-concave if $a_i^2 \geq a_{i-1}a_{i+1}$.*

This property gets its name from the fact that if $\{a_i\}_{i=1}^n$ is log-concave, then the sequence $\{\log a_k\}_{i=1}^n$ is concave. The connection between unimodality and log-concavity is given by the following proposition.

**Proposition 1.** *A log-concave sequence is unimodal.*

*Proof.* Rearranging the defining inequality for log-concavity, we see that

$$\frac{a_i}{a_{i-1}} \geq \frac{a_{i+1}}{a_i},$$

so the ratio of consecutive terms is decreasing. Until the ratios decrease below 1, the sequence is increasing, and after this point, the sequence is decreasing, so it is unimodal.    □

Given the definition of log-concavity, it is straightforward to design a regularization term that encourages log-concavity:

$$R = \sum_{i=2}^{n-1} \max\{0, a_{i-1}a_{i+1} - a_i^2\}. \tag{11}$$

By Proposition 1, this regularizer also encourages unimodality.

## B  MORE DATASETS AND IMPLEMENTATION DETAILS

### B.1  MORE DETAILS ON THE DATASET AND EXPERIMENTAL SETUP

**HMDB51 and UCF101** The dataset pre-processing and data augumentation are the same as the ResNet ImageNet experiment (He et al., 2016). All the videos are resized to $224 \times 224$ resolution and fed into a ResNet-50 pretrained on ImageNet. The last convolutional layer feature map size is $2048 \times 7 \times 7$. The experimental setup for the Moments in Time dataset is the same as HMDB51 and UCF101 except time sequence and image resolution.

**Moments in Time** For the Moments in Time dataset (Monfort et al., 2018), the videos only have 3 seconds, much shorter than HMDB51 and UCF101. We extract RGB frames from the raw videos at 5 fps. Therefore, the sequence length is $T = 15$. Following the practice in (Monfort et al., 2018) to make all the videos uniform resolution, we resize the RGB frames to $340 \times 256$ pixels. When extracting features, we use the ResNet-50 pretrained on ImageNet model using resized images with resolution of $256 \times 256$ pixels. The data augmentation is the same as the ResNet ImageNet experiment (He et al., 2016). The feature map size of the last convolutional layer is $2048 \times 8 \times 8$.

**THUMOS14** The action detection task of THUMOS'14 (Jiang et al., 2014) consists of 20 classes of sports activities, and contains 2765 trimmed videos for training, while 200 and 213 untrimmed videos for validation and test respectively. Following the standard practice (Yeung et al., 2016; Zhao et al., 2017), we use the validation set as training and evaluate on the testing set. Following the standard practice (Xu et al., 2017) to avoid the training ambiguity, we remove the videos with multiple labels. We extract RGB frames from the raw videos at 10 fps. The last convolutional layer feature map size is $2048 \times 7 \times 7$.

### B.2  SPATIAL ATTENTION NETWORK ARCHITECTURE

The detailed architecture of spatial attention network described in Section 3.2 is listed in the Table 5.

| Index | Inputs | Operation | Output shape |
|:-----:|:------:|:---------:|:------------:|
| (1) | - | $X_i$ | $2048 \times H \times W$ |
| (2) | (1) | CONV-(N1024, K3, S1, P1), BN, ReLU | $1024 \times H \times W$ |
| (3) | (2) | CONV-(N512, K3, S1, P1), BN, ReLU | $512 \times H \times W$ |
| (4) | (3) | CONV-(N1, K3, S1, P1), Sigmoid | $1 \times H \times W$ |

Table 5: Architecture of spatial attention network. $H$ and $W$ are the height and width of the feature map, respectively.

### B.3  MORE IMPLEMENTATION DETAILS

All the experiments are evaluated on machines with a single Nvidia GeForce GTX 1080Ti GPU. The networks are implemented using the Pytorch library and our code will be publicly available with the paper.

## C  MORE RESULTS

### C.1  MORE SPATIAL TEMPORAL ATTENTION RESULTS

Fig. 7 shows more results on spatial temporal attention.

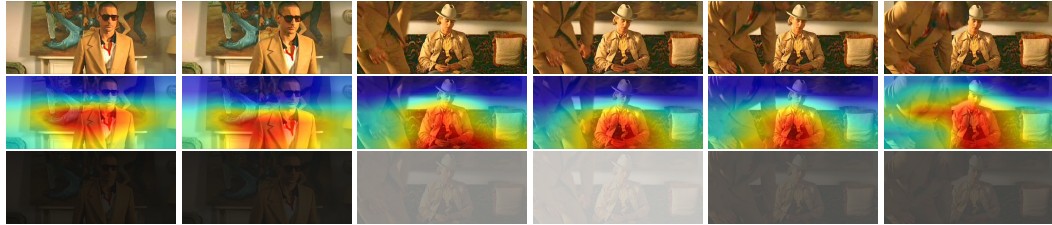

Figure 7: **Multiple actions in one image for video action recognition** The *Sit* action from HMDB51. In the first two frames, there is no sitting action while the spatial attention capture the important area, but the temporal attention can effectively ignore them as the background information. It is interesting that in the last few frames, there is another person trying to sit down, but the visual attention can only capture one sitting person.

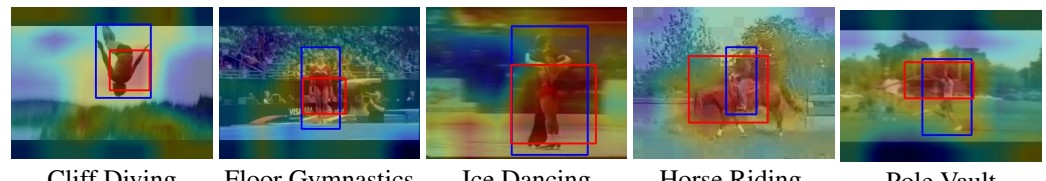

| Cliff Diving | Floor Gymnastics | Ice Dancing | Horse Riding | Pole Vault |

Figure 8: **Examples of spatial attention for action localization.** (Best viewed in color.) Blue bounding boxes represent ground truth while the red ones are predictions from our learned spatial attention. Our spatial attention mechanism is able to focus on important part of the action, while the ground truth bounding boxes labels focus on the entire human pose. As in the training stage, the ground truth bounding boxes are not used, and the model can only depend on crucial spatial area rather than the entire action to make prediction. For actions with object interactions, such as *Horse Riding* and *Pole Vault*, the ground truth box focuses on human pose while the model focuses on objects (such as Horse, Pole) as well.

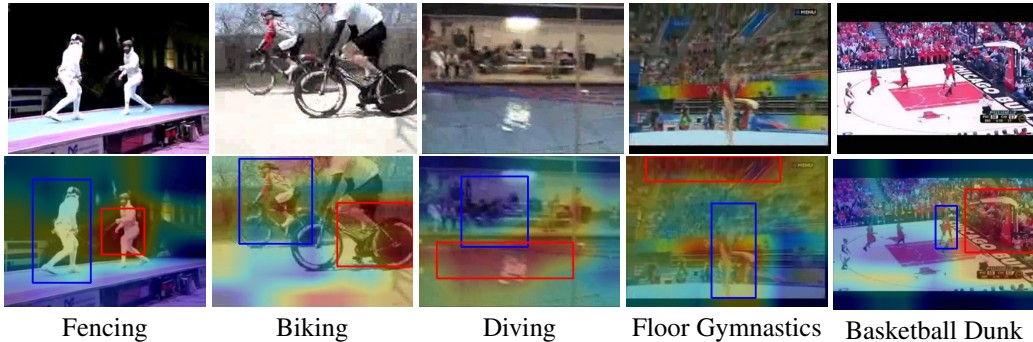

| Fencing | Biking | Diving | Floor Gymnastics | Basketball Dunk |

Figure 9: **Failure cases for spatial localization.** (Best viewed in color.) In the ground truth bounding boxes, there is only one bounding box in human action for each frame but there may be more than one person performing the same action. Typical IOU=0 case is that our attention focuses on the unlabeled human action, such as *Fencing* and *Biking* shown here. Strong motion blur also leads to failure cases, such as the *Diving* and *Floor Gymnastics* frames shown here. The diving and gymnastics poses are highly motion blurred so the spatial attention focuses on the swimming pool and audiences respectively. Some of the important background information also leads to failure cases, such as the basketball frame in the *Basketball Dunk* shown here.

## C.2 MORE SPATIAL LOCALIZATION RESULTS

Fig. 8 shows more spatial localization results. Fig. 9 shows some failure cases.

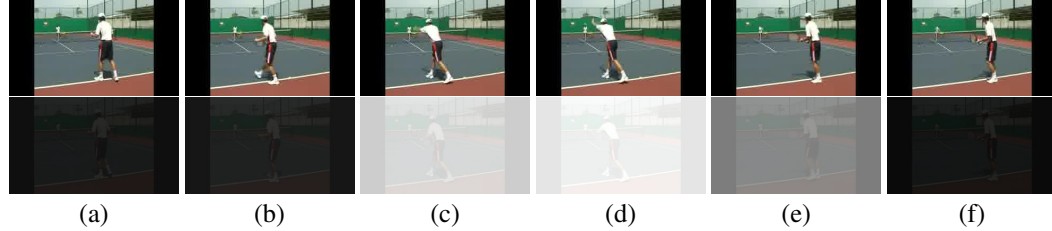

| (a) | (b) | (c) | (d) | (e) | (f) |

Figure 10: **Failure example for temporal attention localization.** A sequence of *Tennis Swing* action from one video of THUMOS14. All temporal localization is correctly localized except the frame (b). The labeled action starts from frame (b) but our temporal attention module still assigns a low importance score.

| Base network | ResNet50 | ResNet101 | ResNet152 |
|---|---|---|---|
| w/o spatial-temporal attention | 47.5 | 49.7 | 50.1 |
| w our spatial-temporal attention | **49.8** | **53.1** | **54.4** |

Table 6: **Top-1 accuracy (%) on HMDB51 with different base networks**

## C.3 MORE ACTION RECOGNITION RESULTS

Table 6 shows results of our spatial-temporal attention model with different base networks. Our spatial-temporal attention mechanism is a easy plug-in model which could be based on different network architectures, and can boost performance.

## C.4 MORE TEMPORAL LOCALIZATION RESULTS

Fig. 10 shows more results on temporal localization with our temporal attention.

