# OpenReview forum: "Where and when to look? Spatial-temporal attention for action recognition in videos"
_ICLR.cc/2019/Conference_

### Official Review · AnonReviewer2 · 2018-11-02
**A spatial-temporal attention model, missing some baselines.**

**Rating:** 6
**Confidence:** 4

**Review:**

The paper propose an end-to-end technique that applies both spatial and temporal attention. The spatial attention is done by training a mask-filter, while the temporal-attention use a soft-attention mechanism.  In addition the authors propose several regularization terms  to directly improve attention. The evaluated datasets are action recognition datasets, such as HMDB51, UCF10, Moments in Time, THUMOS’14. The paper reports SOTA on all three datasets.



Strengths:

The paper is well written: easy to follow, and describe the importance of spatial-temporal attention.

The model is simple, and propose novel attention regularization terms.

The authors evaluates on several tasks, and shows good qualitative behavior.


Weaknesses:

The reported number on UCF101 and HMDB51 are confusing/misleading.  Even with only RGB, the evaluation miss numbers of models like ActionVLAD with 50% on HMDB51 or Res3D with 88% on UCF101. I’ll also add that there are available models nowadays that achieve over 94% accuracy on UCF101, and over 72% on  HMDB51. The paper should at least have better discussion on those years of progress. The mis-information also continues in THUMOS14, for instance R-C3D beats the proposed model.

In my opinion the paper should include a flow variant. It is a common setup in action recognition, and a good model should take advantage of these features. Especially for spatial-temporal attention, e.g., VideoLSTM paper by Li.

In general spatial attention over each frame is extremely demanding. The original image features are now multiplied by 49 factor, this is more demanding in terms of memory consumption than the flow features they chose to ignore.  The authors reports on 15-frames datasets for those short videos. But it will be interesting to see if the model is still useable on longer videos, for instance on Charades dataset.

Can you please explain why you chose a regularized making instead of Soft-attention for spatial attention?

To conclude:
The goal of spatial-temporal attention is important, and the proposed approach behaves well. Yet the model is an extension of known techniques for image attention, which are not trivial to apply on long-videos with many frames. Evaluating only on rgb features is not enough for an action recognition model. Importantly, even when considering only rgb models, the paper still missed many popular stronger baselines.

---

> ### Author Response · Authors · 2018-11-26
> **Reply to Reviewer3: add more attentional model baselines. VideoLSTM is full supervised method and we are weakly supervised method**
>
> We would like to thank the reviewer for the detailed comments and suggestions for the manuscript. We have updated the paper and highlight the major changes with red colour. The following replies are used to address the concerns.
>
> 1. So sorry for the confusing numbers in HMDB51/UCF101. We have already updated the results and add more baselines in Table 1 of the paper, please refer to the updated version.  For these two datasets, we mainly used them as ablation study and compare with state-the-art attention based methods with the same RGB inputs, rather than competing the state-of-the-art accuracy with other highly complicated and computational expensive models, such as I3D.  For HMDB51/UCF101, our baselines are other state-of-the-art attention models (such as papers from NIPS2017 [1] and CVIU2018 [2]) with RGB images and RGB images using the same base network without attention mechanism. We achieve better results than baseline methods.
>
> 2. For the spatial localization on Thumos 14 dataset, we have added the results of R-C3D [3] as one of our baseline method in our updated version. But notable to say that R-C3D is a full supervised learning method which needs bounding boxes during training, which is very expensive as bounding boxes are needed for each frame. While our method is a weakly supervised learning method which is trained only with class labels but without bounding boxes.
>
> 3. The spatial attention is just simply using several layers of convolution, which is not very computationally demanding. For UCF101 and HMDB51, we also tested on a smaller number of frames, the results are very similar when using 25 and 50 frames. Many the video action recognition literatures, for instance, attentional pooling[1], videoLSTM[2], and  the two-stream network[4],  use 25 frames of RGB images. I think the bottleneck of using optical flow for large datasets is the optical flow extraction time and storage, especially for large datasets, such as Moments in Time dataset.  If using 25 RGB frames, motion stream also use 25 optical flow frames.
>
> 4. For the comparison with VideoLSTM [2],
> (1) The spatial-temporal attention mechanisms are different: they used LSTM for spatial attention, while we are using ConvLSTM for temporal attention. We also use different mechanisms for spatial attention: they are using ConvLSTM for spatial attention, while we are using several layers of ConvNet to learn an attention mask.
> (2) For temporal attention, [2] means impose similar visual attention to motion stream with optical flow input. But our temporal attention uses different mechanism with RGB frames as input.
> (3) For localization, the VideoLSTM can only do spatial localization, our spatial-temporal attention can do both spatial and temporal localization.
> (4) Compared with their RGB stream results, our results are much better and please refer to Table 1 of our newly updated paper.
>
> [1] Girdhar, Rohit, and Deva Ramanan. "Attentional pooling for action recognition." Advances in Neural Information Processing Systems (NIPS). 2017.
> [2] Li, Zhenyang, et al. "VideoLSTM convolves, attends and flows for action recognition." Computer Vision and Image Understanding (CVIU). 2018.
> [3] Xu, Huijuan, Abir Das, and Kate Saenko. "R-C3D: region convolutional 3d network for temporal activity detection." IEEE Int. Conf. on Computer Vision (ICCV). 2017.
> [4] Simonyan, Karen, and Andrew Zisserman. "Two-stream convolutional networks for action recognition in videos." Advances in neural information processing systems (NIPS). 2014.

---

> > ### Comment · AnonReviewer2 · 2018-11-27
> > **Although simple, model does seem to be useful**
> >
> > After adding more results, and perform a better comparison, I have a stronger feeling that their model is contributing to the current literature in video action recognition. The novelty is still limited, because spatial-temporal attention was discussed in video-lstm, and if we go out of the action recognition task, papers like "TGIF-QA: Toward Spatio-Temporal Reasoning in Visual Question Answering" discussed it as well. But I agree their approach is simpler than the LSTM based attention in VideoLSTM. The ablation study also suggest that spatial attention, as well as their regularization terms are beneficial.
> >
> > I'm still curious if you have tried a naive spatial soft-attention (i.e., use a softmax over a learned spatial scores).
> >
> > To conclude, given the better comparison, I tend to recommend acceptance.

---

### Official Review · AnonReviewer1 · 2018-11-03
**Minor novelty**

**Rating:** 3
**Confidence:** 5

**Review:**

A method for activity recognition in videos is presented, which uses spatial soft attention combined with temporal soft attention. In a nutshell, a pixelwise mask is output and elementwise combined with feature maps for spatial attention, and temporal attention is a distribution over frames. The method is tested on several datasets.

My biggest concern with the paper is novelty, which is rather low. Attention models are one of the most highly impactful discoveries in deep learning, which have been widely and extensively studied in computer vision, and also in activity recognition. Spatial and temporal attention mechanisms are now widely used by the community. I am not sure to see the exact novelty of the proposed, it seems to be very classic: soft attention over feature maps and frames is not new. Using attention distributions for localization has also been shown in the past.

This also shows in the related works section, which contains only 3 references for spatial attention and only 2 references for temporal attention out of a vast body of known work.

The unimodality prior (implemented as log concave prior) is interesting, but uni-modality is a very strong assumption. While it could be argued that spurior attention should be avoided, unimodality is much less clear. For this reason, the prior should be compared with even simpler priors, like total variation over time (similar to what has been done over space).

The ablation study in the experimental section shows, that the different mechanisms only marginally contribute to the performance of the method: +0.7p on HMDB51, slightly more on UCF101. Similarly, the different loss functions only very marginally contribute to the performance.

The method is only compared to Sharma 2015 on these datasets, which starts to be dated and is not state of the art anymore. Activity recognition has recently very much benefitted from optimization of convolutional backbones, like I3D and variants.

The LSTM equations at the end of page are unnecessary because widely known.

---

> ### Author Response · Authors · 2018-11-26
> **Reply to Reviewer2: address our novelty, add more state-of-the-art baselines and re-run several experiments**
>
> We would like to thank the reviewer for the detailed comments and suggestions on the manuscript.  We have updated the paper and highlight the major changes with red colour. The following replies are used to address the concerns.
>
> 1. Novelty:
> We agree with the reviewer that the attention model has been widely used in many different tasks (please refer to our introduction and related work part). The novelty of each work usually lies on using different attention mechanisms and their applications are for different tasks. Our focus is on attention mechanism for video action recognition. Different from the LSTM based soft-attention is only used for spatial localization [1, 2, 3], but we used it for temporal localization. For spatial attention, we propose a new method which just uses several convolutional layers to learn an attention mask, which is novel and has not been used before. We also introduced two regularizers for spatial attention. For temporal attention, we introduce the convolutional LSTM based mechanism and a regularizer. This temporal mechanism also considers the salient spatial information which learned by our spatial attention in the previous step.
>
> 2.  The purpose of the unimodality prior is fundamentally different from the total variation (TV) regularization. The TV regularization encourages attention weights to remain the same in consecutive frames. However, it does not encourage sparsity and it is not sufficient on the action recognition dataset.  On the contrary, we found empirical evidence that salient information in most videos is contained only in a few consecutive frames. Therefore it is sensible to make the unimodality assumption on the importance of frames.
>
> 3. We added two state-of-the-art visual attention video action recognition baselines [2][3] which also use attention mechanism with the RGB images as input. Our results are better than these two methods. We did not compare with I3D and variants as they are using 3D convolutions and too computationally expensive. The attention model is not a totally independent model, but a plug-in model. The performance of the entire action recognition network not only depends on the attention model, but also the backbone model.
> For instance, the accuracy for the base network ResNet50 is 47.78%, with our spatial-temporal attention, the accuracy is 49.93%. For ResNet101, the base network accuracy is 49.73%, with our spatial-temporal attention, the accuracy achieves  53.07%. For ResNet152, the base network accuracy is 50.04%, with our spatial-temporal attention, the accuracy achieves 54.44%.
> The I3D network is very computationally expensive and data hungry, and need to pretrain on large datasets, such as Kinetics. Currently we are not using I3D network due to computational limitations, as pretraining needs 8 GPUs and train 2 weeks. Our attention model could extend to spatial-temporal attention based on 3D networks by learning a 3D spatial mask and the frame temporal attention with Convolutional LSTM. It will be an interesting direction if computational resources are limited.
>
> [1] Sharma, Shikhar, Ryan Kiros, and Ruslan Salakhutdinov. "Action recognition using visual attention." arXiv preprint arXiv:1511.04119 (2015).
> [2] Li, Zhenyang, et al. "VideoLSTM convolves, attends and flows for action recognition." Computer Vision and Image Understanding (CVIU), 2018.
> [3] Girdhar, Rohit, and Deva Ramanan. "Attentional pooling for action recognition." Advances in Neural Information Processing Systems (NIPS). 2017.

---

> > ### Comment · AnonReviewer1 · 2018-11-26
> > **Minor novelty**
> >
> > I thank the authors for the answers.
> > I still think that the novelty of this paper is too minor to be considered for publication.

---

### Official Review · AnonReviewer3 · 2018-11-04
**Nice and diverse experiments, slightly limited novelty**

**Rating:** 6
**Confidence:** 4

**Review:**

# 1. Summary
This paper presents a novel spatio-temporal attention mechanism. The spatial attention is decomposed from the temporal attention and acts on each frame independently, while the temporal attention is applied on top of it on the temporal domain. The main contribution of the paper is the introduction of regularisers that improve performance and interpretability of the model.

Strengths:
* Quality of the paper, although some points need to clarified and expanded a bit more (see #2)
* Nice diversity of experiments, datasets and tasks that the method is tested on (see #4)

Weaknesses:
* The paper do not present substantial novelty compared to previous work (see #3)


# 2. Clarity and Motivation
The paper is in general clear and well motivated, however there are few points that need to be improved:
* How is the importance mask (Eq. 1) is defined? The authors said “we simply use three convolutional layers to learn the importance mask”, however the convolutional output should be somehow processed to get out the importance map, in order to match the same sizes of X_i. The details of this network are missing to be able to reproduce the model.
* The authors introduced \phi(H) and \phi(X) which are feedforward networks, but their definition and specifics are not mentioned in the paper.
* It is not clear how Eq. 9 performs regularization of the mask. Can the authors give an intuition about the definition of L_{contrast}? What does it encourages? In which cases might it be useful?
* Why does L_{unimodal} need to encourage the temporal attention weights to be unimodal? It seems that the assumption is valid because of the nature of the dataset, i.e., the video clips contain only a single action with some “background” frames in the beginning and the end. This is not valid in general. Can the authors discuss about this maybe with an example?


# 3. Novelty
The main concern of the proposal in this paper is its novelty. Temporal attention pooling have been explored in other papers; just to cite a popular one among others:
* Long, Xiang, et al. "Attention clusters: Purely attention based local feature integration for video classification." Proceedings of the IEEE Conference on Computer Vision and Pattern Recognition. 2018.
* Other paper from the youtube8m workshops explore the same ideas: https://research.google.com/youtube8m/workshop2017/
Sec. 2.2 should be expanded by including papers and discuss how the presented temporal attention differs from that.

Moreover spatio-temporal attention has been previously explored. For example, the following paper also decouple the spatial and temporal component as the proposal:
* Song, Sijie, et al. "An End-to-End Spatio-Temporal Attention Model for Human Action Recognition from Skeleton Data." AAAI. Vol. 1. No. 2. 2017.
This is just an example, but there are there are other papers that model the spatio-temporal extent of videos without attention for action recognition. The authors should expand Sec. 2 by including such relevant literature.


# 4. Experimentation
The experiments are carried on video action recognition task on three public available datasets, including HMDB51, UCF101 and Moments in Time. The authors show a nice ablation study by removing the main components of the proposed method and show nice improvements with respect to some baseline (Table 1). Although the results are not too close to the state of the art for video action recognition on HMDB51 and UCF101, the authors first show nice accuracy on Moments in Time (Table 2).

Moreover the authors show that the model can be useful on the more challenging task of weakly supervised action localization (UCF101-24, THUMOS). Specifically, spatial attention is used to localize the action in each frame by thresholding, showing competitive results (Table 3). Although some more recent references are missing, see the following paper for example:
* G. Singh, S Saha, M. Sapienza, P. H. S. Torr and F Cuzzolin. "Online Real time Multiple Spatiotemporal Action Localisation and Prediction." ICCV, 2017.
Then the authors tested also for temporal action localization (Table 4).

In general, the paper is not showing state-of-the-art results, however the diversity of experiments, datasets and tasks that are presented makes it pretty solid and interesting.

---

> ### Author Response · Authors · 2018-11-26
> **Reply to Reivewer1: more explanations on our novelty**
>
> We would like to thank the reviewer1 for the detailed comments and suggestions for the manuscript. We have updated the paper and highlight the major changes with red color.
>
> 1: Novelty
> (1) comparison with [1]:
> We use a different attention mechanism. For temporal attention, we use attention mechanism based on Convolutional LSTM, which to our knowledge is novel and has not been used before. In contrast, [1] uses cluster attention.
> The temporal cue in [1] assumes that temporal information is not important.  I think it depends on the dataset, for some dataset, shuffling/reversing the image sequence may have very small or no influence on the final performance, for instance, something-something dataset is highly influenced by temporal information while UCF101 is less influenced by shuffling [3]. Kinetics is also influenced by temporal ordering, the performance drops a lot when reversing the image sequences [4]. At least temporal information has never been proved harmful for the video action recognition.
> The result reported in [1] for HMDB51 and UCF101 are obtained using a model with both optical flow and RGB streams. Our method does not use the optical flow stream, therefore, we did not compare with this method directly.
> (2) Comparison with Youtube 8M: Although our temporal attention is also based on LSTM, the convolutional LSTM used in our work is different from the Youtube8M workshop, and our temporal attention also considered the spatial information, especially the spatial attention which learned from our previous spatial attention module.
> (3) We expanded the related work section in the revised and updated version of the paper. Changes include adding [8] and other related works mentioned by the reviewer.
>
> 2. Clarity and Motivation
> (1) Importance mask: we listed the detailed network structure for the importance mask in appendix  B.2 in the newly updated version.
>  \phi(H) and \phi(W) are two fully-connected networks used for generating temporal attention weights, the input \phi(H_{t-1}) is the hidden layer feature map, and the input \phi (X{t}) is the current feature map.
> (2) The contrast loss is to make the action foreground and background separable for attention map. According to Eq.(9), the first term encourages the mask value of the foreground region (M_i>0.5) to be 1, and the second term encourages the mask value of background region (M_i<0.5) to be 0.
> (3) We agree with the reviewer that because of the nature of dataset as currently all the datasets we are using contain only a single action and usually this action happens in a sequence of frames. Here we do not consider a video which has more than one action class as we only have one label for each video. We agree with the reviewer that each video has one label have some limitations, it will be interesting to explore one video with multiple labels for future work.
>
> 3. Experiments
> (1). Comparison with previous work [5] for temporal localization:
> It is important to highlight that our method is weakly supervised, i.e., only classification labels are used during training. In other words, no temporal labels are used. While [5] is a full supervised method. Therefore, we did not compare with this method but only compare with the reinforcement learning based method [6] and weakly supervised method [7] which share our setting and inputs.
>
> (2) The performance of video action recognition attention model depends on both the attention model and the backbone architecture. A stronger backbone will generally improve the performance of our method.
> For instance, if the base network is ResNet50, the accuracy is 47.78% without attention; with our spatial-temporal attention, the acc is 49.93%.  We also run new experiments using base network ResNet101 and ResNet152.  For ResNet101, the base network accuracy is 49.73%; with our spatial-temporal attention, the accuracy reaches  53.07%. For ResNet152, the base network accuracy is 50.04%; with our spatial-temporal attention, the accuracy reaches 54.44%.
>
> [1] Attention clusters: Purely attention based local feature integration for video classification." CVPR 2018.
> [2] Other paper from the youtube8m workshops explore the same ideas: https://research.google.com/youtube8m/workshop2017/
> [3] Zhou, Bolei, Alex Andonian, and Antonio Torralba. "Temporal relational reasoning in videos." ECCV, 2018.
> [4] Xie, Saining, et al. "Rethinking spatiotemporal feature learning: Speed-accuracy trade-offs in video classification." ECCV. 2018.
> [5] G. Singh, S Saha, M. Sapienza, P. H. S. Torr and F Cuzzolin. "Online Real time Multiple Spatiotemporal Action Localisation and Prediction." ICCV, 2017.
> [6] Yeung, Serena, et al. "End-to-end learning of action detection from frame glimpses in videos." CVPR. 2016.
> [7] Wang, Limin, et al. "Untrimmednets for weakly supervised action recognition and detection." ICCV. 2017.
> [8] Song, Sijie, et al. "An End-to-End Spatio-Temporal Attention Model for Human Action Recognition from Skeleton Data." AAAI. 2017.

---

> > ### Comment · AnonReviewer3 · 2018-11-28
> > **Comment to authors related to novelty**
> >
> > I thank the authors to expand more on my comments/questions.
> >
> > I understand that the implementation of attention in this paper is a bit different than others, however the underling idea is the same, which is why the novelty of the model is not major.

---

### Public Comment · ~Yaser_Souri1 · 2018-12-03
**Contrast Loss**

If someone just reads your paper from section 3, you never mention how $M_i$ is computed. For example whether a sigmoid is applied or not.
Here I assume a sigmoid is applied.

Now regarding the contrast loss:
There is no gradient flowing from $B_i$ back in your implementation. Am I right?

Also have you considered the addition of Gaussian noise before the sigmoid to $M_i$s as a way to make the masks more contrasty? I saw this idea in [1].

[1]: Online and Linear-Time Attention by Enforcing Monotonic Alignments
Colin Raffel, Minh-Thang Luong, Peter J. Liu, Ron J. Weiss, and Douglas Eck
34th International Conference on Machine Learning (ICML), 2017.

---

### Meta-Review · Area_Chair1 · 2018-12-13
**Solid work but novelty concerns held back the paper from rising above acceptance threshold**

**Confidence:** 4
**Recommendation:** Reject

**Metareview:**

Strengths: The paper presentation was assessed as being of high quality. Experiments were diverse in terms of datasets and tasks.

Weaknesses: Multiple reviewers commented that the paper does not present substantial novelty compared to previous work.

Contention: One reviewer holding out on giving a stronger rating to the paper due to the issue of novelty.

Consensus: Final scores were two 6s one 3.

This work has merit, but the degree of concern over the level of novelty leads to an aggregate rating that is too low to justify acceptance. Authors are encourage to re-submit to another venue.